# Modular DNA strand-displacement controllers for directing material expansion

Joshua Fern [1] & Rebecca Schulman [1,2]

Soft materials that swell or change shape in response to external stimuli show extensive promise in regenerative medicine, targeted therapeutics, and soft robotics. Generally, a stimulus for shape change must interact directly with the material, limiting the types of stimuli that may be used and necessitating high stimulus concentrations. Here, we show how DNA strand-displacement controllers within hydrogels can mediate size change by interpreting, amplifying, and integrating stimuli and releasing signals that direct the response. These controllers tune the time scale and degree of DNA-crosslinked hydrogel swelling and can actuate dramatic material size change in response to <100 nM of a specific biomolecular input. Controllers can also direct swelling in response to small molecules or perform logic. The integration of these stimuli-responsive materials with biomolecular circuits is a major step towards autonomous soft robotic systems in which sensing and actuation are implemented by biomolecular reaction networks.

---

[1] Chemical and Biomolecular Engineering, Johns Hopkins University, Baltimore, MD 21218, USA. [2] Computer Science, Johns Hopkins University, Baltimore, MD 21218, USA. Correspondence and requests for materials should be addressed to R.S. (email: rschulm3@jhu.edu)

To survive in diverse or changing conditions, cells and tissues adapt their shape or behavior in response to a broad array of biochemical stimuli[1]. Molecular receptors detect stimuli and trigger signal transduction pathways that amplify and integrate incoming chemical information and then direct a response. The separation of the cellular machinery for sensing, signal processing, and actuation enable these complex responses. Integrating information about multiple inputs allows cells to respond intelligently to complex input combinations, while amplification of input signals makes it possible for cells to trigger a response in which high concentrations of molecules must be activated or transformed, such as in cell migration, differentiation, or growth[2–4]. This organization means that new cell functions can be developed through the rewiring of signal transduction pathways to reconnect or add inputs and responses[5]. Analogous systems for signal processing within engineered materials could likewise allow materials to emulate the complex responsiveness of cells. Soft materials, hydrogels in particular, where molecular systems for sensing and actuation could operate in aqueous medium, are especially amenable to this approach.

Stimuli-responsive hydrogels offer important advantages over traditional materials and can be used as membranes[6,7] or in tissue engineering[8]. Hydrogels that respond to a variety of biomolecules, including enzymes[9–11], antibodies[12], nucleic acids[13], or combinations of molecular inputs[14–16] have been developed. However, because these stimulus molecules must interact directly with the hydrogel network, a new material must be developed for each new stimulus. Physiologically, for example, detecting each of the thousands to tens of thousands of proteins[17] and RNA/small molecules that convey important information about the local environment quickly becomes infeasible using such an approach. Direct interaction between the stimulus and material also sets a particular concentration threshold for a stimulus to direct a response, which may or may not correspond to the desired sensitivity. In vivo, the nominal concentration of biological molecules, depending on the target molecule, ranges from sub-picomolar to millimolar[18,19].

Recently, chemical reaction networks that amplify low-concentrations of input signals have been shown to direct gelation[6,11,14,15], signal propagation[20], or gel–sol transitions[6,21]. Here we show that we can build modular material controllers that combine amplification with logic, translation of input signals, and response tuning to directly and precisely program dramatic material size change. Hydrogel size change is critical for polymer actuation or shape change processes that drive self-folding materials or soft robots[22], and for mediating delivery of therapeutic materials such as nanoparticles[23].

Here we develop molecular controllers that sense different low-concentration inputs, process and amplify those inputs, and then direct large-scale responses in hydrogel materials. Within our system, a programmable chemical controller decides whether to produce an output signal that is then amplified to produce a high-concentration actuation signal. This signal directs the material to use a separate supply of chemical fuel to induce size change (Fig. 1a). The system we develop thus has a modular controller, actuator, and energy source to program mechanical work, making it a primitive soft robot in which all parts are implemented as biochemical processes.

## Results

### DNA-crosslinked hydrogels as state-switchable devices. We started with a DNA-crosslinked polyacrylamide hydrogel as the material substrate (Fig. 1b). DNA-crosslinked hydrogels can respond to temperature[24,25], ions[25,26], nucleic acids[13], and small molecules[27–30] by de-hybridizing or changing the persistence

lengths of the crosslinks, leading to size and shape change or changes in elastic modulus. Recently, it was shown that DNA hairpins incorporating into the crosslinks could trigger up to 100-fold volumetric expansion of DNA-crosslinked hydrogels (Fig. 1c)[31]. We hypothesized that upstream DNA circuits such as amplifiers[32–34], translators[35,36], or logic circuits[37,38] could release DNA outputs that might direct this expansion. These circuits could in turn take different types or concentrations of chemical stimuli as inputs.

We modified the hydrogel crosslinks so that they could be either in an active state, where DNA hairpins can direct hydrogel expansion, or an inactive state, where crosslinks are unable to interact with hairpins (Fig. 1d). Crosslinks were designed to switch from an inactive to an active state via a toehold-mediated DNA strand-displacement process[39,40]. The strand that unlocks the crosslink is a short DNA strand of a form that can be the output of DNA strand-displacement logic[37,38] or sensing[35,36,41] circuits. We designed the locking strand to minimize leaky swelling of locked gels in an iterative process outlined in Supplementary Note 1 and Supplementary Figs. 1–6.

### DNA-crosslinked hydrogels as a model swelling system. We characterized swelling kinetics using hydrogel spheres synthesized in a droplet-based photopolymerization process we developed (see Methods and Fig. 2a). Spheres expand evenly, such that swelling kinetics can be reliably measured by measuring changes in their radius or area. We measured the area of each particle's 2D projection in fluorescence micrographs and used this value as the metric for particle size (Supplementary Note 2, Supplementary Fig. 7). The average radius of the synthesized particles after equilibration in buffer was $570 \pm 7\,\mu m$ (95% confidence interval from standard deviation, $n = 196$, see Methods and Supplementary Note 3). The projections of 93.4% of particles were at least 90% circular (Supplementary Fig. 8).

We first verified that hydrogel particles synthesized with active crosslinks (i.e., without locks) swell in the presence of their corresponding DNA fuel, a mixture of polymerizing and terminating monomers (Fig. 1c, Supplementary Fig. 9). Both polymerizing and terminating monomers can insert into crosslinks; after incorporation, polymerizing monomers present a site where subsequent monomers can insert, while terminating monomers do not[31]. Particles swelled at a roughly constant initial rate, then more slowly as they approached a final size (Fig. 2b). The fluorescence intensity of the hydrogels decreased as their sizes increased due to decreasing rhodamine density (Fig. 2b, Supplementary Fig. 7). The intensity of the particles also became non-uniform during swelling, suggesting hydrogel swelling began near the particle surface, then proceeded in the center (Fig. 2b). The swelling rate and final size could be tuned across a wide range by adjusting the monomer concentration or the fraction of monomers that were terminating, consistent with earlier studies (Fig. 2b, Supplementary Fig. 10)[31]. To compare the influence of different locks and controllers on a standard swelling process in subsequent experiments, we used $20\,\mu M$ of each monomer, of which 10% were terminators. A lower fraction of terminators produced more swelling but was slower to reach final size and more difficult to track in a single field of view on a microscope (Fig. 2b).

### Activating particles with Key strands. When hydrogel particles with locked crosslinks (Figs. 1d, 3a) were incubated with hairpin fuel, only a $3.7 \pm 4.5\%$ change in area was observed over 60 h, as compared to $260 \pm 2\%$ for crosslinks without locks (Fig. 3b, Supplementary Fig. 11). Most swelling of locked particles occurred in the first 5 h. Increasing the hairpin concentration to

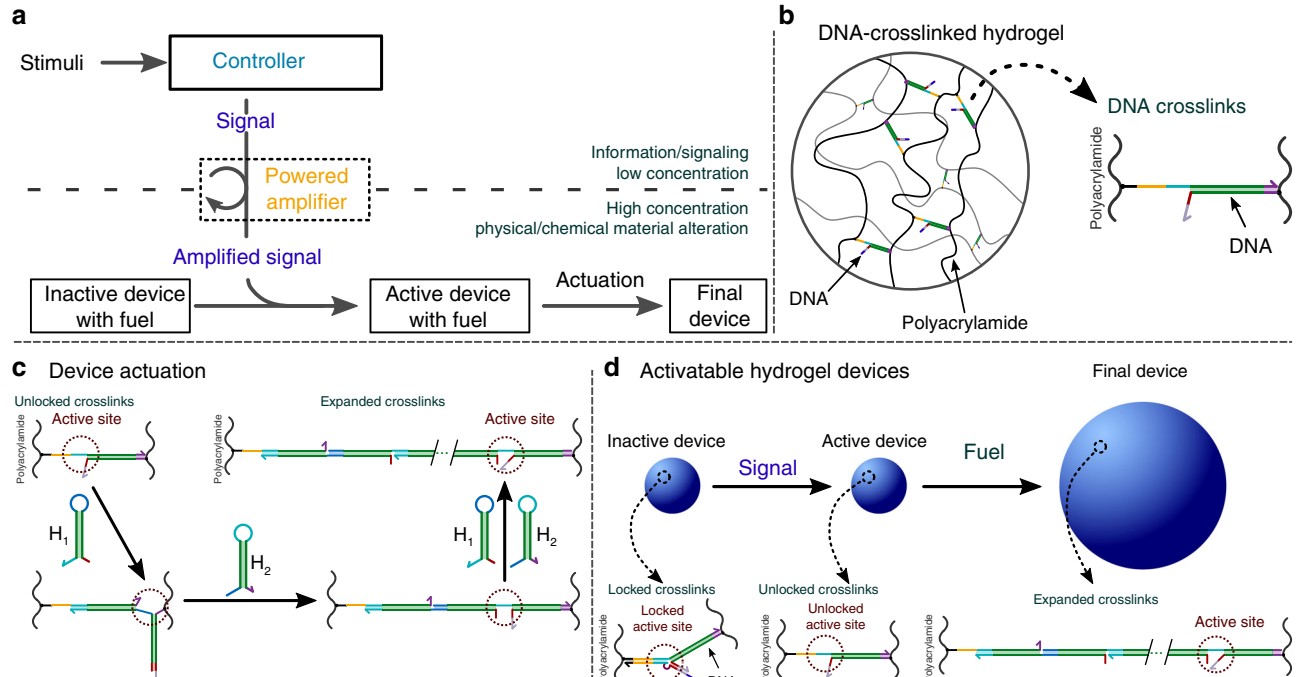

**Fig. 1** Scheme for controlled expansion of DNA-crosslinked hydrogels gated by biomolecular signals. Colors within DNA strands show sequence domains, drawn to scale; complementary regions are shown in the same colors. **a** Device architecture. A controller interprets stimuli and decides whether to switch the device from an inactive to an active state. The controller takes low-concentration inputs and processes them with low-concentration molecular circuit elements; the output is then amplified to activate the high-concentration crosslinks within the device that interact with high-concentration fuel to drive material shape change. **b** In a DNA-crosslinked hydrogel, hybridized DNA oligonucleotides crosslink polyacrylamide chains. **c** Schematic of the hairpin monomer incorporation process that drives size change. Two types of hairpin monomers are inserted into crosslinks in alternating series at a single active site. **d** Control over whether size changes is achieved by control over device activation through crosslink unlocking. Unlocked crosslinks can extend by incorporating hairpin fuel

200 μM did not further increase the degree locked particles swelled (Supplementary Fig. 12). Thus, hairpin fuel does not trigger significant expansion of a hydrogel with locked crosslinks.

We next tested whether adding a Key strand that can unlock crosslinks (Fig. 3a) could allow the particles to expand in the presence of fuel. We first incubated the locked particles with fuel for 24 h to allow the fuel monomers to diffuse into the particles, then added different concentrations of Key strand to different particles. Key strands induced swelling (Fig. 3b) and particles swelled more in the presence of higher concentrations of Key strand (Fig. 3c). This is consistent with increases in swelling observed with increasing fractions of active (vs. inactive) crosslinks (Supplementary Fig. 13).

**Activating particles through catalytic crosslink unlocking**. We next asked how we might couple the unlocking process to biomolecular circuits, allowing inputs other than nucleic acid signals or combinations of inputs to trigger material shape change. However, the concentrations of Key strand required to trigger swelling are higher than the 1 nM to 1 μM typical for the outputs of biomolecular circuits[20,32–35,37,38,42–48].

We thus asked whether we could induce swelling in response to lower concentrations of a trigger molecule. Using such a system would also mean that smaller concentrations of biomolecular circuit components could be used, making it much more practical and cost-efficient to implement complex information processing or control systems[37,45]. Such a system could also allow amplification of molecular signals that are present at concentrations lower than 10 μM to induce a response, expanding the range of sensitivity of shape-change materials.

We thus designed a molecular amplification process to allow one input molecule to unlock many crosslinks. We based our design on catalytic DNA strand-displacement circuits[32,33,39,44,45,49,50] where an input strand first triggers the release of an output and is then released by a helper molecule that is consumed in the reaction. These catalytic circuits can amplify the input signal 100–100,000-fold[32–34,39,49]. With this approach, only the hairpins and the molecules responsible for fueling the amplification circuit would be required to be at high concentrations, while the trigger could be at a significantly lower concentration.

In the catalytic crosslink unlocking process we designed, the Key strand is replaced by Catalyst and Helper strands (Fig. 4a). The Catalyst strand can bind to a locked crosslink and partially release the lock. The Helper strand can then bind to the Catalyst-crosslink complex, producing an unlocked crosslink and a waste complex, while the Catalyst is released to unlock another crosslink. Thus, in the presence of a larger Helper concentration, a small concentration of Catalyst should activate the device. We designed the Helper and Catalyst so as to minimize undesired crosslink unlocking in the absence of Catalyst and maximize the amount of crosslink unlocking in the presence of both Helper and Catalyst (Supplementary Note 1).

Without the Helper strand, a Catalyst strand should still be able to unlock one crosslink, but the Catalyst will not be released after unlocking (Fig. 4a). Therefore, in the absence of Helper, the Catalyst should work like the Key strand and produce roughly the same amount of swelling at the same concentrations. Indeed, we observed that 10 μM of Catalyst was needed to achieve the same high-degree change in area of 270 ± 1% over 60 h as seen in response to 10 μM of Key strand. Over the same time period, <2%

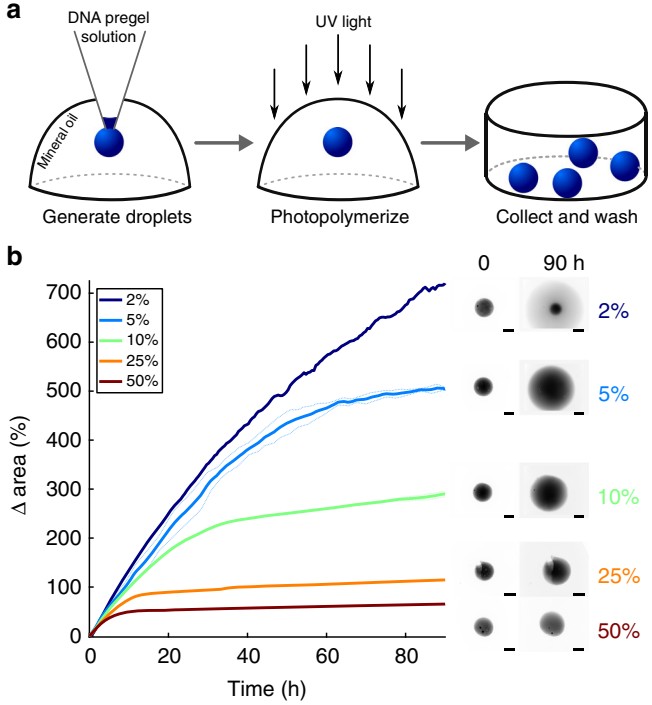

**Fig. 2** Hydrogel particle synthesis and tunable swelling with DNA hairpin fuel and terminators. **a** Hydrogel particles are prepared by pipetting droplets of pre-polymer acrylamide-DNA crosslinker solution into mineral oil and polymerizing the droplets with UV light. Rhodamine-B is incorporated into the acrylamide backbone to allow fluorescence imaging of particles (see Methods). **b** The changes in area of the 2D projection of particles without locks over time for different fractions of hairpin monomers that are terminating hairpins. Each solid curve represents 1 particle (2%, 25%, 50%) or the average of measurements of 2 particles (5%, 10%); dotted lines are the individual replicates of each average. The edges of particles expanded with 2% terminator monomer were unable to be discerned from the surrounding solution after 90 h. Hairpin fuel with 10% terminator was chosen for further hydrogel swelling experiments because particles with 2% and 5% terminator expanded to a diameter larger than the microscope's field of view. Image intensities are scaled based on each image's minimum and maximum intensity. Scale bars: 500 μm

increase in area was observed at Catalyst concentrations of 100 nM or less (Supplementary Fig. 14).

Because there is no toehold where the Helper can bind to the locked crosslink to initiate fast displacement of the crosslink lock, little to no unlocking (and thus expansion) should occur in the presence of Helper strand and fuel but no Catalyst. As expected, locked hydrogel particles incubated with Helper strands and fuel expanded just 8 ± 2% after 40 h at Helper concentrations of 1 μM or below (Supplementary Fig. 15). At 10 μM of Helper strand, 47 ± 4% expansion was observed after 40 h, which is significant but still well below the 200–250% increase in area observed in response to 10 μM of Key or Catalyst over the same time period (Fig. 3c, Supplementary Fig. 14).

In contrast, when as little as 100 nM Catalyst was added to particles with 10 μM Helper and hairpin fuel, particles expanded as fast as particles without locks (Fig. 4b). Larger concentrations of Catalyst strand did not significantly increase the extent of swelling, consistent with the idea that Catalyst strands are each capable of initiating multiple unlocking reactions. The high concentration of Helper is also critical to expansion: particles expanded 5-fold less in the first 24 h in response to 100 nM Catalyst and 1 μM Helper than in response to 100 nM Catalyst and 10 μM Helper (Fig. 4b, Supplementary Figs. 16–17). At high

Helper strand concentrations, the concentration of Catalyst controlled the rate of swelling (Supplementary Fig. 17). This observation is consistent with the idea that when Catalyst concentrations are small, significant Catalyst turnover is required before most crosslinks have been unlocked.

**Small molecule triggered controllers of hydrogel size change.** We next asked whether we could couple molecular circuits to the catalytic unlocking process by designing DNA strand-displacement circuits that produced the Catalyst strand as an output. We first designed an aptasensor circuit that releases a strand containing the Catalyst sequence only when ATP is present (Fig. 5a, Supplementary Fig. 18)[35].

In the absence of ATP, the Catalyst sequence is partially sequestered in double-stranded form, preventing it from interacting with the hydrogel crosslinks. A Cofactor strand can bind the ATP sensor complex to partially separate the two strands of the complex. The ATP-bound sensor can then displace the Catalyst from the Catalyst source complex[35]. This system of interactions ensures that Catalyst can be released at a significant rate only when ATP and the Cofactor are both present in solution.

We characterized the release of Catalyst in response to ATP in solution using a fluorophore-quencher reporting assay (Methods, Supplementary Fig. 19). In the presence of 100 nM each of the ATP sensor complex, Catalyst source complex, and Cofactor, ATP triggered Catalyst release at a rate dependent upon ATP concentration (Fig. 5b). Catalyst concentrations of about 75 nM, the smallest concentration we observed that produced fast particle swelling, were produced only in response to ATP concentrations above 500 μM, while less than 40 nM Catalyst was produced at 100 μM ATP. When no ATP was present, some Catalyst was still produced, possibly due to the ability of a Cofactor-ATP sensor complex to react slowly with the Catalyst source complex even in the absence of ATP. The need for hundreds of micromolar ATP concentrations to produce tens of nanomolar of Catalyst is consistent with the $K_d$ of the original aptamer-ATP interaction $(1–10 \mu M)$[51] and the expected increase in $K_d$ caused by the required modifications that enable the aptamer's binding-induced structure-switching that provides an input to strand-displacement circuits.

When locked hydrogel particles were incubated with the ATP-driven controller circuit, the amount of particle expansion depended on ATP concentration (Fig. 5c). Interestingly, the dose-response relationship between ATP concentration and particle swelling was somewhat digital. ATP concentrations below 500 μM did not significantly increase the rate of swelling over the baseline rate observed in response to 0 μM ATP, whereas swelling rates were similar for ATP concentrations of 500 μM and above. This behavior can be understood by composing the responses of the controller and the circuit: the controller releases Catalyst concentrations that produce similar expansion rates for ATP concentrations above 500 μM and for ATP concentrations below 500 μM. Supporting this interpretation, we found that the controller, the catalytic unlocking process, and expansion process appear to be modular processes--the addition of the ATP-sensing circuit (but no input) to the amplifier circuit, crosslink locks, and hairpin fuel did not change the swelling rate (Supplementary Fig. 20). This modularity suggests that one could replace either the aptamer sensor or the crosslinks to trigger hydrogel expansion in response to different chemical inputs or to direct expansion of hydrogels with different crosslink sequences. However, because unintended interactions can occur at different concentrations of the DNA species (Supplementary Fig. 21), designing and tuning the system as a whole to reduce such interactions may be required.

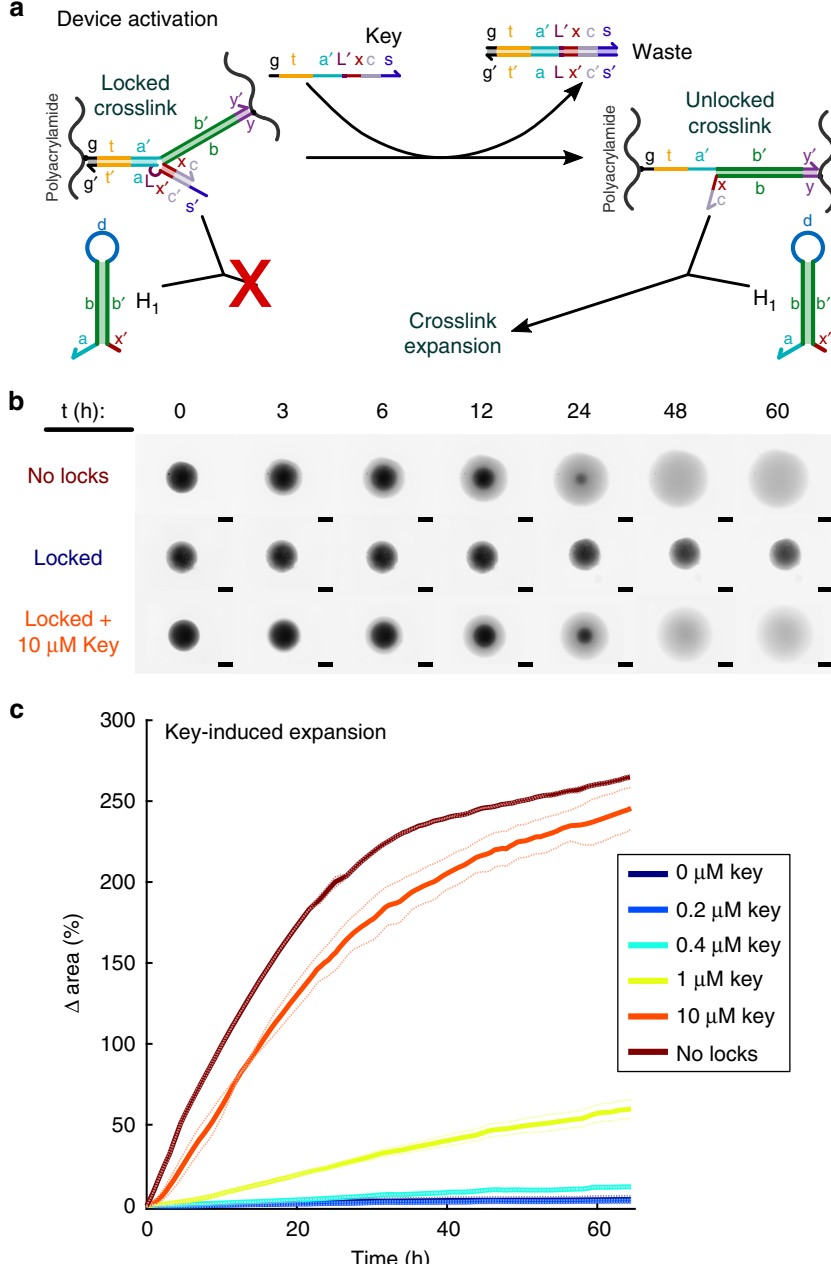

**Fig. 3** Hydrogel particle unlocking. **a** Schematic of crosslink unlocking by a Key strand through strand-displacement. **b** Fluorescence micrographs of particles. Particles without locks swell when hairpin fuel is added, but particles with locked crosslinks show less than 4% change in area over 70 h (Supplementary Fig. 11). Addition of Key strands to locked particles allows swelling. Scale bars: 500 μm. **c** The change in area of the 2D projection of each particle over time for different concentrations of Key strand added at time 0. Solid curves are the averages of measurements of 2 particles; dotted lines are the individual replicates of each average

**Hydrogel actuation triggered by combinations of inputs**. We next tested whether hydrogel expansion could be directed in response to specific combinations of multiple inputs, each presented at small concentrations. Previously, hydrogels have been engineered to change color, gel, or swell slightly in response to logical combinations of inputs[14,15,27]. However, in these systems, the inputs interacted directly with the crosslinks, limiting the range of potential chemical inputs and necessitating very high input concentrations (μM–mM) to elicit the response. Our controller design circumvents these limitations. The controller can interpret input signals that do not interact with the material and direct in situ signal amplification within the controller, making it possible to direct changes in response to low (100-200 nM) input concentrations.

We modified a previously developed DNA strand-displacement AND logic circuit[38] to release concentrations of Catalyst strand sufficient to trigger hydrogel expansion only when both inputs are present (Fig. 5d–e, Supplementary Fig. 22). As designed, significant particle swelling occurred only in the presence of both inputs (Fig. 5f). Without inputs, the addition of the circuit components did not increase the swelling rate (Supplementary Fig. 23), demonstrating that, like the ATP-sensing controller, this controller operates modularly with respect to the amplification and unlocking processes. Some swelling was observed in response

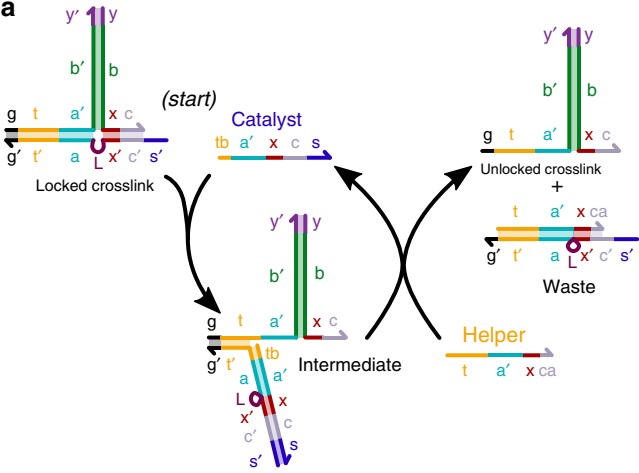

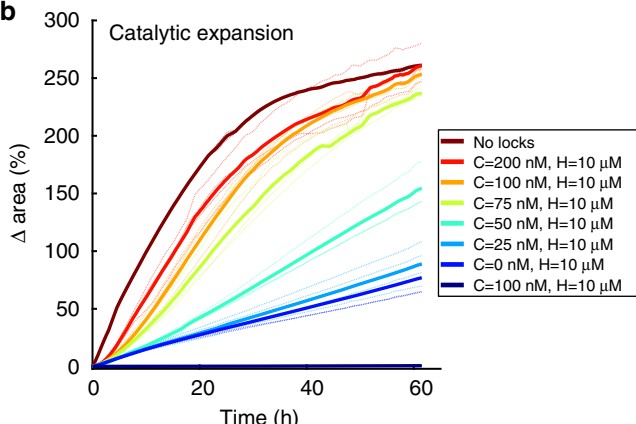

**Fig. 4** Catalytic crosslink unlocking enables a signal at concentrations of 100 nM or less to trigger fast, high-degree swelling. **a** A Catalyst strand unlocks a crosslink via toehold-mediated strand-displacement to form an intermediate unlocked crosslink complex. A Helper strand reacts with this intermediate to release the Catalyst strand, allowing it to unlock another crosslink. **b** Expansion in response to different concentrations of Catalyst and 10 μM Helper strands added to locked particles pre-incubated with DNA hairpins. Helper strands unlock crosslinks without Catalyst, but much more slowly than with Catalyst. Solid curves are the averages of measurements of 2–4 particles; dotted lines are the individual replicates for each average

to input $F_{in}$ alone (Supplementary Fig. 23a), perhaps due to sequence similarity between $F_{in}$ and the Catalyst, notably at toehold domains $s$ and $cb$ (Supplementary Fig. 22). To better illustrate how logic can gate large-scale material change, we organized movies of particles in the presence of different input combinations expanding into a swelling "truth table" (Supplementary Movie 1).

Interestingly, while the swelling behavior observed is digital, the controller does not contain a nonlinear threshold amplifier typically required for digital logic: the concentration of the output should simply be the minimum of the concentrations of the inputs. Digital behavior is observed because the catalytic expansion process performs the required nonlinear transformation. If the controller's output is above about 75 nM, catalytic amplification induces fast swelling, whereas for lower catalyst concentrations very little swelling occurs. Because DNA strand–displacement amplifiers, like the catalytic amplifier, can produce output even in the off state due to undesired "leak" interactions between system components[33,37], the ability to

operate without one likely improves the controller's reliability. This design also suggests how modular circuits coupled to material systems can exploit the behavior of the material itself for control to maximize both performance and system simplicity.

## Discussion

Here we have shown how to use catalytic amplification of a small concentration of a trigger molecule to direct a dramatic change in material size, demonstrating systematically that engineered signaling processes between species at low concentrations can control the chemistry and behavior of dense materials which contain orders of magnitude more material and mass that must be transformed than the stimulus or the circuit (Supplementary Note 4). This system allows tens of nanomolar of an input signal to change the conformations of material components present at millimolar concentrations.

Because the inputs to the controllers do not interact directly with the hydrogel polymer, it is straightforward to create components where different stimuli could induce the same response. While it will be important to characterize interactions among the complex system of fuel, catalytic, and controller molecules we have created (e.g., Supplementary Fig. 24), and to better understand what physical properties control the extent of hydrogel swelling in response to changes in crosslink structure[31], the modular design of our system suggests that there is no immediate barrier to increasing the complexity of the circuits that direct the response. Controllers could now readily be built that employ further amplification[34], control expansion timing[42], respond to single-base pair changes in inputs[50], or direct phase transitions[52]. The expansion demonstrated in this paper can direct material shape change[25,31]. Multiplexed circuits that each control the activity of crosslinks in a single material domain (Supplementary Figs. 13, 25) or different material domains[31] could orchestrate complex shape change tasks. This work thus shows how many of the fundamental mechanisms employed in robots for open-loop control could be transported readily to aqueous biomolecular material systems. In combination with sensors, it might also be possible to develop chemo-mechanical feedback processes for closed loop control.

## Methods

**Chemicals and DNA.** Acrylamide (Bio-Rad, Cat. No. 161-0100) was solubilized using MilliQ purified water. Rhodamine B-conjugated methacrylate monomer was obtained from PolySciences, Inc (Cat. No. 25404-100) and used for fluorescent visualization of hydrogels. Hydrogels were polymerized using the photoactive initiator Irgacure 2100 (BASF). ATP was purchased from Sigma (Cat. No. A6419) and solubilized to 53 mM using MilliQ purified water. Unmodified and acrydite-modified DNA strands were purchased with standard desalting purification from Integrated DNA Technologies, Inc. Fluorophore- and quencher-modified DNA was purchased with HPLC purification. All DNA was solubilized using TAE buffer (Life Technologies, Cat. No. 24710-030) supplemented with 12.5 mM magnesium acetate tetrahydrate (Sigma, Cat. No. M5661). As described in Supplementary Figs. 2, 9, 18, and 22, DNA sequences were designed using NUPACK (Supplementary Methods)[53] or adapted from previous literature[31,37,38,54]. Sequences used in this study are found in Supplementary Table 1.

**Preparation of DNA complexes.** DNA complexes were annealed in TAE buffer supplemented with 12.5 mM magnesium acetate (TAEM) from 90 to 20 °C using an Eppendorf PCR at 1 °C/min. Hydrogel crosslink complexes were annealed at a stock concentration of 3 mM per strand while all other complexes were annealed at 100 μM. Hairpin-forming strands were flash cooled on ice for 3 min after heating to 95 °C for 10 min at a concentration of 80 μM or 200 μM. Hairpin and crosslink complexes were not further purified. All other multi-strand circuit components (e.g., Source complexes) were PAGE purified after annealing using 15% polyacrylamide gels at 150 V for 3–4.5 h. Immediately prior to PAGE purification, all complexes, with the exception of the ATP sensor complex, were incubated ~16–20 h with 50 μM of their respective input strand with the toehold removed (see Supplementary Table 1 for sequences)[38]. Fluorophore-/quencher-modified DNA complexes (Reporters) were not PAGE purified after annealing at 50 μM.

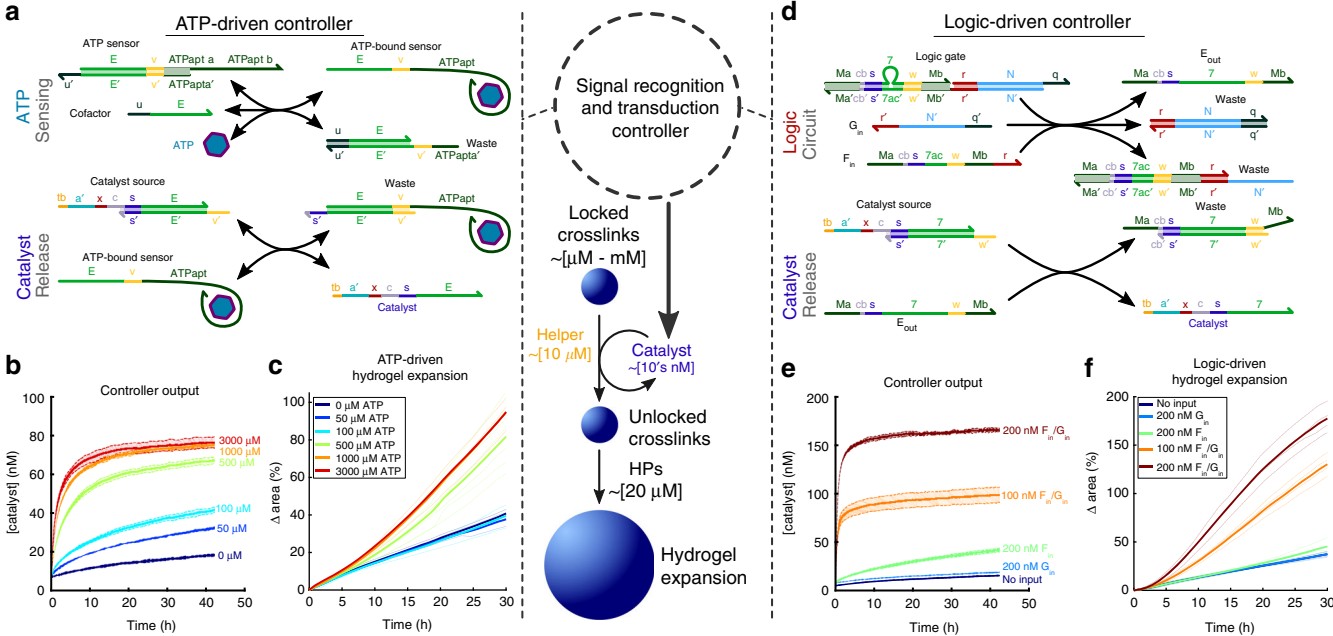

**Fig. 5** Controllers that direct hydrogel expansion in response to different types of chemical inputs. **a** An aptamer-containing strand-displacement circuit that releases Catalyst in response to ATP. **b** Solution kinetics of the circuit in (**a**) measured using fluorescence reporting (see Methods, Supplementary Fig. 19) in response to different ATP concentrations. ATP sensor, Cofactor, and Catalyst source initially at 100 nM. **c** ATP-controlled rates of particle swelling mediated by the circuit in (**a**), catalytic crosslink unlocking (Fig. 3a), and hairpin polymerization (Fig. 1c). Sensor, Cofactor, and Catalyst source at 100 nM; Helper strand at 10 μM. **d** DNA strand-displacement AND logic circuit. Two inputs, $F_{in}$ and $G_{in}$, are required for Catalyst release. **e** Kinetics of the logic circuit in (**d**) in solution measured by fluorescence reporting. The logic gate and Catalyst source are at 200 nM. **f** Particle expansion in the presence of different $F_{in}$ and $G_{in}$ concentrations. Logic gate and Catalyst source are at 200 nM; Helper at 10 μM. $N = 3$ (**b**, **e**) or 2–6 **c**, **f**. Shaded regions (**b**, **e**) represent 95% confidence intervals determined by standard deviations; dotted lines in (**c**, **f**) are replicate measurements with the averages represented by solid lines

**Synthesis of poly(DNA-co-acrylamide) hydrogel particles**. DNA crosslinks were mixed to a final concentration of 1.154 mM with water, ×10 TAEM, acrylamide, rhodamine methacrylate, and Irgacure 2100 (75% v/v in butanol). The final concentrations of acrylamide, rhodamine methacrylate, and Irgacure 2100 were 1.41 M, 2.74 mM, and 3% (v/v), respectively. After mixing, the pre-polymer solutions were degassed under vacuum for 5 min. Pre-polymer droplets were prepared using a water-in-oil method (Fig. 2a). Mineral oil USP (CVS Pharmacy) "wells" were prepared on a cratered parafilm surface and pre-polymer droplets were added using a pipette set to 0.25 μL. Droplets were exposed to 365 nm light using a Benchtop 3UV Transilluminator (UVP) for 1 min (~4 mW/cm²) to polymerize and crosslink the particles. Particles were purified from the oil using centrifugation into TAEM and were stored at 4 °C until use, usually within 1 week.

**Swelling of DNA-crosslinked hydrogels**. Swelling experiments were conducted in 96-well plates (Fisher Scientific) with one particle per well. Micrographs of particles were taken on an IX73 Olympus fluorescence microscope using a rhodamine filter. The final volume of liquid in each well varied between 100–120 μL, depending on the experiment. For experiments with locked particles, the particles were incubated with DNA hairpins (20 μM/hairpin type, 10% terminator) for about 24 h prior to the addition of Catalyst/Helper strands or circuit complexes. For all experiments with DNA circuits, the Helper strand concentration was 10 μM. Images of each particle were captured every 30 min.

**Particle area measurement and analysis**. Images of the fluorescent particles were considered to be accurate 2D projections of the particle size near the center xy-plane. To decrease the sensitivity and bias involved in measuring the diameter, especially of an irregular or non-circular projection, the area of the 2D projection was chosen as the representative variable of particle size and calculated as a function of time for each particle. The area was determined using standard intensity-based thresholding and mask image analysis using MATLAB (Supplementary Note 2). Area measurements for each particle were normalized to the initial time point. The curves showing the change in size as a function of time are taken from measurements made every 30 min, averaged over multiple particles. The curves were smoothed with a window size of 3.

**Fluorophore-quencher assay of DNA controller circuits**. An Agilent Stratagene Mx3000 or Mx3005 was used to test the operation of the DNA-based circuits in the absence of hydrogel particles. A reporter complex, using FAM and IowaBlackFQ fluorophore-/quencher-modified DNA, was designed to increase measured

fluorescence upon reaction with DNA strands containing the Catalyst sequence and toehold (Supplementary Fig. 19). The measured fluorescence increase was converted into the concentration of Catalyst strand using a calibration curve. The DNA strand-displacement logic circuit was tested using 200 nM Source complexes and 200 nM Reporter. Aptasensor circuits were tested at 100 nM Source complexes, 100 nM Cofactor strand, and 200 nM Reporter. PolyT₂₀ (1 μM) was added to inhibit adsorption to well walls.

**Code availability**. The programming code that was used to analyze the raw data that supports the findings of this study are available from the corresponding author upon request.

## Data availability
The data that support the findings of this study are available from the corresponding author upon reasonable request.

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

## Acknowledgements

We thank D. Scalise, A. Cangialosi, and S. Schaffter for thoughtful discussions. This research was supported by Department of Energy Early Career Award 221874 to R.S.

## Author contributions

J.F. and R.S. designed experiments. J.F. conducted experiments and carried out experimental analysis. All authors discussed the results and wrote the manuscript.

## Additional information

**Competing interests:** The authors are pursuing a patent application for this work.

