## [Peer Review File · Nature Communications]

Reviewers' Comments:

Reviewer #1:

Remarks to the Author:

The paper "Modular DNA Strand-Displacement Controllers for Directing Material Expansion" by Fern and Schulman describes the results from a beautiful set of experiments showing that a variety of chemical controllers (implemented using DNA) can modulate the swelling of hydrogels. They develop a hierarchy of interesting implementations, ranging from swelling only in the presence of a Key molecule, to a catalytic system which can detect and respond to much smaller input concentrations, and finally to controllers that sense other types of molecules (ATP instead of DNA) or execute AND logic (i.e. both of two inputs must be present to trigger swelling.) The development of new and programmable soft materials is of fundamental and practical interest, and the experiments described herein are carefully done and thoroughly interpreted. Furthermore, this paper bridges together ideas different communities, like chemomechanics and DNA nanotechnology, and is likely to be of broad interest. Thus I recommend that the paper be published following minor revisions, which are outlined below.

Questions/comments/revisions:

- A general comment: A more detailed discussion of the sequence design would strengthen the paper and encourage others to build on the neat experiments demonstrated here. How was the NUPACK design tool used? What were the selection criteria? Which molecular details are most important? Are thermodynamics or kinetics dominant? Are details like the toehold lengths relevant? And so on.
- A key feature of this work is that small concentrations of inputs can be amplified to large concentrations needed to trigger significant volume changes, and that this might allow these materials to respond to "physiological concentrations" (line 40 for example). Can the authors be more specific? Can they provide ballpark concentrations of example sensory inputs?
- The definition of "projected area" and the algorithm by which it is computed is a bit confusing. It would be helpful if the authors expanded their discussion of how it is calculated and showed the projected areas along with the micrographs in Figure 2.
- The demonstration that the swelling rate and final size can be controlled by adjusting the concentration and proportion of polymerizing/terminating monomers is interesting and useful. Perhaps the authors should move these results from the SI to the body of the manuscript.
- The authors make the comment that "The ability to induce swelling in response to low concentrations of a trigger molecule would also mean that smaller concentrations of the molecular circuit components could also be used, making it much more practical to implement complex processing systems requiring many different species," but this conclusion is not obvious. The authors should explain their claim more fully if this is a key advantage of their methods. How would this work? Doesn't any system still require a large concentration of "Helper" strands to unlock the crosslinks?
- The authors find that the Helper alone can trigger expansion of the gel when added at high concentrations (10 μ M). Do they have a hypothesis for why this happens or how it might be designed out?
- The ATP-driven controller is a fantastic idea. However, the efficiency of the controller in the current implementation is pretty low. Can the authors comment on why there's a 1000-fold reduction in the concentration of the Catalyst for a given ATP concentration (i.e. 500 μ M ATP produces 75 nM catalyst)? How might this be improved?

Reviewer #2:

Remarks to the Author:

The manuscript by Schulman et. al. showed to build modular material controllers that combine amplification with logic, translation of input signals, and response tuning directly and precisely program dramatic material size change. This controller shows how to use a small concentration of a trigger molecule to direct a dramatic change in material size. Meanwhile, the inputs can be also designed with small molecule and logic gate, which further expand the applications of this modular DNA strand displacement controller.

I have two concerns regarding the DNA hydrogels controlled by modular DNA strand displacement reaction. 1. It's very interesting to demonstrate as low as 100-200 nM input concentration to induce direct changes of soft hydrogel size expansion, but any lower input concentration has been tested to induce apparent size change? 2. Since authors claim this controller's perspective as a primitive soft robot to be implemented as biochemical process, has the performance of DNA hydrogel controller been tested under biological environment, at least under cellular lysate? If not, please discuss the potential difficulties applied in biological complexity and what'll be the potential solutions?

Reviewer #3:

Remarks to the Author:

The paper describes the integration of programmable, DNA-based molecular computing components with actuatable hydrogels. These have been previously shown to undergo triggered swelling based on a form of hybridization chain reaction that inserts hairpin monomers into a growing DNA chain that forces apart the hydrogel polymers chains at the points where the hydrogel crosslinks with the DNA.

The key development in this paper is the integration of a "locking" mechanism into the initially crosslinker, so that the swelling process cannot commence without a particular "key" strand unlocking the toehold for the fuel hairpin monomers to attach to. This then allows the use of DNA-based catalytic cycles to carry out the unlocking reaction in a multiple turnover manner, so that the key is recycled after it has unlocked a crosslink and can then proceed to unlock more crosslinks. This provides an elegant solution to an issue raised by the authors' previous work on this topic, which was that high concentrations of the trigger strands were required to cause hydrogel expansion. The authors demonstrate that equally strong responses can be obtained with a fraction of the input concentration.

Furthermore, this abstraction of the unlocking procedure behind a DNA strand displacement reaction allows it to be wired up to a range of different inputs in a modular fashion. This is demonstrated by implementing a two-input DNA strand displacement "AND" logic gate that triggers expansion only if both input strands are present, as well as an ATP-sensitive aptamer gate that released the required unlocking strand due to a competitive interaction with an ATP ligand-binding process. These are compelling demonstrations of the power of DNA-based molecular logic as a flexible, universal framework for sensing a range of different inputs and connecting them to an actuation process.

The paper is very well written and enjoyable to read, and the results are highly compelling. The results are novel and are likely to be of significant interest to both the molecular computing and biomaterials communities. As such I recommend accepting the paper.

One minor comment I would raise is that there do not appear to be any control experiments that demonstrate the sequence specificity of the interactions between the crosslinkers and the

key/catalyst etc. Given the sequence-programmable nature of DNA strand displacement I am willing to believe that they would work as expected, but it would be a powerful demonstration that multiplexed processes could be integrated into a single material, e.g., to direct different responses depending on which stimuli are observed.

Dear Reviewers,

Herein we address each reviewer's comments point by point. For clarity, the reviewer's comments are italicized and our responses are unformatted.

Reviewer #1 (Remarks to the Author):

The paper "Modular DNA Strand-Displacement Controllers for Directing Material Expansion" by Fern and Schulman describes the results from a beautiful set of experiments showing that a variety of chemical controllers (implemented using DNA) can modulate the swelling of hydrogels. They develop a hierarchy of interesting implementations, ranging from swelling only in the presence of a Key molecule, to a catalytic system which can detect and respond to much smaller input concentrations, and finally to controllers that sense other types of molecules (ATP instead of DNA) or execute AND logic (i.e. both of two inputs must be present to trigger swelling.) The development of new and programmable soft materials is of fundamental and practical interest, and the experiments described herein are carefully done and thoroughly interpreted. Furthermore, this paper bridges together ideas different communities, like chemomechanics and DNA nanotechnology, and is likely to be of broad interest. Thus I recommend that the paper be published following minor revisions, which are outlined below.

Questions/comments/revisions:

- A general comment: A more detailed discussion of the sequence design would strengthen the paper and encourage others to build on the neat experiments demonstrated here. How was the NUPACK design tool used? What were the selection criteria? Which molecular details are most important? Are thermodynamics or kinetics dominant? Are details like the toehold lengths relevant? And so on.

As per the reviewer's recommendation, we have added the section labeled Supplementary Methods 1 that provides more detail into the design of the sequences using NUPACK. The majority of the sequences were taken from previous literature, but additional sequences were designed for the locking/unlocking mechanism and parts of the controllers. These additional sequences in the controllers were designed using a 3-DNA base alphabet, an established method for preventing undesired interactions between groups of strands. We used NUPACK simulations to verify that there was minimal or no unintended hybridization between the crosslinks, locking mechanism, and the pre-existing sequences at thermodynamic equilibrium. Due to the multiple potential timescales of dynamic processes in this system (including those for crosslink unlocking, hairpin insertion, and hydrogel swelling), it is difficult to determine whether

achieving a desired thermodynamic equilibrium or whether avoiding undesired kinetics during the strand-displacement reactions are most important for design. We chose 4 base pairs as the toehold length for the Catalyst binding to the locked crosslink, which is expected to determine the order of magnitude of the reaction rate constant. The toehold for the Helper strand to bind to the Catalyst-locked crosslink intermediate complex was chosen to be 1 base pair longer than the Catalyst binding toehold. The criteria for these designs was based upon previous characterizations of toehold-mediated strand-displacement and catalytic circuit designs.¹⁻⁵

- A key feature of this work is that small concentrations of inputs can be amplified to large concentrations needed to trigger significant volume changes, and that this might allow these materials to respond to “physiological concentrations” (line 40 for example). Can the authors be more specific? Can they provide ballpark concentrations of example sensory inputs?

We thank the reviewer for their comment regarding input concentrations that are physiologically relevant, and we have tried to clarify this point in the text. Different physiologically relevant molecules are present at sub-picomolar to millimolar concentrations either inside the cell or within the extracellular space. For example, glucose in the blood is around 7 mM.⁶ Inside cells, metabolite concentrations can be present over a very wide range – from sub-picomolar to tens of millimolar.⁸ Thus, the key point is not that there is a universal range for all molecules but that there are concentration ranges of different signals that convey information. Our goal is to make it possible to control what ranges of a signal create a response, even when that concentration is below the concentration needed for a stimulus without amplification.

Of interest to this manuscript, ATP is at millimolar concentrations within cells and at micromolar concentrations extracellularly.⁹ While these species may be at relatively high or low concentration compared to the 100 nM Catalyst strand we use as a trigger, the controllers or molecular translators that can convert these species into a nucleic acids, *e.g.*, the sensitivity of the aptamer upstream of the Catalyst, controls the amount of signal needed to induce a response. For example, the ATP aptasensor that we used within the manuscript converts micro- to millimolar concentrations of ATP into nanomolar concentrations of the Catalyst-containing strand.

- The definition of “projected area” and the algorithm by which it is computed is a bit confusing. It would be helpful if the authors expanded their discussion of how it is calculated and showed the projected areas along with the micrographs in Figure 2.

We agree with the reviewer that the detail we provided could be improved. We have expanded the text in Supplementary Note 2 and Supplementary Fig. 7 to explain the image analysis algorithm in more detail and the method used to convert the area measured from each image into the change in area shown in the figures. Specifically, because the particles are relatively bright compared to the background, we used standard intensity thresholding to find the pixels that belong to the particle in each image. Since we used epifluorescence microscopy to image the 3D spheroid particles, the resulting images taken by the camera are only 2D representations (circles) of those particles, or the “projection” of the fluorescent light onto that focal plane during imaging.

- The demonstration that the swelling rate and final size can be controlled by adjusting the

concentration and proportion of polymerizing/terminating monomers is interesting and useful. Perhaps the authors should move these results from the SI to the body of the manuscript.

We agree and have moved what was Supplementary Fig. 10 into the main text and is now Figure 2b as per the reviewer's suggestion, and rearranged Figures 1-3 accordingly.

- The authors make the comment that "The ability to induce swelling in response to low concentrations of a trigger molecule would also mean that smaller concentrations of the molecular circuit components could also be used, making it much more practical to implement complex processing systems requiring many different species," but this conclusion is not obvious. The authors should explain their claim more fully if this is a key advantage of their methods. How would this work? Doesn't any system still require a large concentration of "Helper" strands to unlock the crosslinks?

We thank the reviewer for their comment on the clarity of that claim. Currently, DNA-based circuits can be built where there are 10's to 100's of DNA strands and complexes interacting with each other to generate an output.¹⁰ Requiring each of these numerous species be present at high concentrations would be prohibitive. Our approach requires only hairpins and the Helper strand be at high concentrations, regardless of the complexity of the controller circuits, thus decreasing the total amount of DNA needed for a circuit by a large factor. We have clarified the sentence in question in the main text and in the following paragraph (in the "Activating particles through catalytic crosslink unlocking" section).

- The authors find that the Helper alone can trigger expansion of the gel when added at high concentrations (10 μM). Do they have a hypothesis for why this happens or how it might be designed out?

We thank the reviewer for this question about the Helper strand being able to trigger expansion. As we have discussed in paragraph 2 of Supplemental Note 1, our hypothesis is that this leak is due to de-hybridization of the bases at the junction of the locking strand with the A-R crosslink complex some fraction of the time. This would provide a small toehold for the Helper strand to initiate crosslink unlocking. The very high concentrations of both the Helper strand and the crosslinks within the gel mean that this reaction could proceed even when the absolute rate is very small; a rough analysis of the rate of the leak interaction suggests a rate constant of unlocking on order $0.1 \text{ M}^{-1} \text{ s}^{-1}$, 6-7 orders of magnitude lower than the rate constant for DNA hybridization and several orders of magnitude slower than the rate constants for the other strand-displacement processes we designed. We reduced the rate of leak by adding unpaired nucleotides to the 3-way junction in the locked crosslinks, but some leak still persists. Design modifications such as including base mismatches for the crosslink within the Helper strand² could be expected to further reduce the leak at the potential expense of reducing the speed of hydrogel unlocking and catalytic turnover.

- The ATP-driven controller is a fantastic idea. However, the efficiency of the controller in the current implementation is pretty low. Can the authors comment on why there's a 1000-fold reduction in the concentration of the Catalyst for a given ATP concentration (i.e. 500 μM ATP

produces 75 nM catalyst)? How might this be improved?

We agree with the reviewer that there is a large difference between the concentration of ATP that is required for the stimulation of swelling and the concentration of Catalyst strand that can induce swelling. However, this difference is expected because the reaction between an aptamer and its ligand is characterized by the K_d of the reaction. When the concentration of the aptamer's substrate is below its K_d , the binding reaction is reverse-biased, which for our cascade means that little ATP-bound sensor is available to release the Catalyst. Since the K_d for the ATP-binding aptamer we used is large, we expect that the concentration of ATP needed to trigger swelling should also be large. We have added a sentence to this effect to the main text. Different aptamer sequences and different desired targets for sensing will have different K_d 's, and thus different levels of Catalyst produced from the same concentration of each aptamer's ligand. The performance of aptamer-based sensing circuits will continue to improve with optimized aptamer design protocols that result in aptamers capable of binding their substrate more tightly, and as new aptamer sequences are released that are capable of the structural reconfiguration upon ligand binding required for integration into strand-displacement circuits.¹¹

Reviewer #2 (Remarks to the Author):

The manuscript by Schulman et. al. showed to build modular material controllers that combine amplification with logic, translation of input signals, and response tuning directly and precisely program dramatic material size change. This controller shows how to use a small concentration of a trigger molecule to direct a dramatic change in material size. Meanwhile, the inputs can be also designed with small molecule and logic gate, which further expand the applications of this modular DNA strand displacement controller.

I have two concerns regarding the DNA hydrogels controlled by modular DNA strand displacement reaction.

1. It's very interesting to demonstrate as low as 100-200 nM input concentration to induce direct changes of soft hydrogel size expansion, but any lower input concentration has been tested to induce apparent size change?

We thank the reviewer for their inquiry into the lower limit of input concentration needed to induce hydrogel expansion. As we have shown in Figure 3 of the main text, we have used input concentrations lower than 100 nM to induce size changes. Specifically, we have shown that 50 and 75 nM of the input Catalyst strand is high enough to induce these dramatic changes. Additionally, the dependence on Catalyst strand concentration that we have shown is consistent with those observed in other DNA strand-displacement amplifier circuits.^{2,3,12} As with those circuit designs, further optimization of our strand-displacement circuit beyond the current degree of optimization (Supplemental Note 1) is required to further reduce the leaks with the Helper strand that we observed, which would enable a decrease in the lower bound of input concentration. As the DNA nanotechnology field further improves the performance of DNA-based amplifiers, we expect that those lessons can be used to improve the design of our circuit. Those future circuits with reduced leak and improved performance could also be coupled to the

amplification circuit we developed here to also enable a reduction in input concentration.¹³ Currently, the ~100 nM trigger that we showed is capable of being generated by the majority of existing DNA strand-displacement and enzymatic circuits,^{1,14-16} enabling the integration of more complex signal processing and molecular translation controllers with the initiation of hydrogel actuation.

2. Since authors claim this controller's perspective as a primitive soft robot to be implemented as biochemical process, has the performance of DNA hydrogel controller been tested under biological environment, at least under cellular lysate? If not, please discuss the potential difficulties applied in biological complexity and what'll be the potential solutions?

We agree that the question of the performance of these hydrogel systems in biological environments is important. DNA-crosslinked hydrogels have been used in cell culture¹⁷⁻¹⁹ and methods that enable DNA strand-displacement circuits to operate in cell culture²⁰ and inside cells²¹ have been and are continuing to be developed. We expect that proper consideration of these techniques would allow our system to be compatible with use with cells in some contexts, but the thorough investigation required to answer those questions does not fit within the scope of this manuscript.

Reviewer #3 (Remarks to the Author):

The paper describes the integration of programmable, DNA-based molecular computing components with actuable hydrogels. These have been previously shown to undergo triggered swelling based on a form of hybridization chain reaction that inserts hairpin monomers into a growing DNA chain that forces apart the hydrogel polymers chains at the points where the hydrogel crosslinks with the DNA.

The key development in this paper is the integration of a "locking" mechanism into the initially crosslinker, so that the swelling process cannot commence without a particular "key" strand unlocking the toehold for the fuel hairpin monomers to attach to. This then allows the use of DNA-based catalytic cycles to carry out the unlocking reaction in a multiple turnover manner, so that the key is recycled after it has unlocked a crosslink and can then proceed to unlock more crosslinks. This provides an elegant solution to an issue raised by the authors' previous work on this topic, which was that high concentrations of the trigger strands were required to cause hydrogel expansion. The authors demonstrate that equally strong responses can be obtained with a fraction of the input concentration.

Furthermore, this abstraction of the unlocking procedure behind a DNA strand displacement reaction allows it to be wired up to a range of different inputs in a modular fashion. This is demonstrated by implementing a two-input DNA strand displacement "AND" logic gate that triggers expansion only if both input strands are present, as well as an ATP-sensitive aptamer gate that released the required unlocking strand due to a competitive interaction with an ATP ligand-binding process. These are compelling demonstrations of the power of DNA-based molecular logic as a flexible, universal framework for sensing a range of different inputs and connecting them to an actuation process.

The paper is very well written and enjoyable to read, and the results are highly compelling. The results are novel and are likely to be of significant interest to both the molecular computing and biomaterials communities. As such I recommend accepting the paper.

One minor comment I would raise is that there do not appear to be any control experiments that demonstrate the sequence specificity of the interactions between the crosslinkers and the key/catalyst etc. Given the sequence-programmable nature of DNA strand displacement I am willing to believe that they would work as expected, but it would be a powerful demonstration that multiplexed processes could be integrated into a single material, e.g., to direct different responses depending on which stimuli are observed.

We thank the reviewer for their recommendation of experiments that demonstrate the sequence-specificity of the unlocking and crosslink extension mechanism. Consistent with previous studies,²² we showed that the swelling process is dependent upon using hairpins with correct and matching sequences as the crosslinks (see “0% Sys1” in Supplementary Fig. 13a, which is 100% System 2 crosslinks incubated with System 1 hairpin fuel). We have clarified this in the figure caption for Supplementary Fig. 13. Supplementary Fig. 12 also demonstrates sequence specificity as per the very minimal swelling observed when hydrogels were incubated with a 20-mer thymine oligonucleotide.

We conducted additional experiments that further demonstrates the sequence specificity of the unlocking process that is included in the revised manuscript as Supplementary Fig. 25. As expected, crosslink unlocking and thus swelling only occurs in the presence of the correct Key strand. Additionally, we showed that this orthogonal set of crosslinks is capable of being catalytically unlocked using Catalyst and Helper strands with the correct sequences for those crosslinks.

In total, our revisions to the manuscript include text and figure changes in the main text and supplemental information. The changes from the original text are highlighted in yellow and we have also provided a new manuscript file without formatting in our revised submission.

We appreciate your continued consideration of our manuscript and look forward to your response.

Sincerely,

Rebecca Schulman
Assistant professor,
Chemical and biomolecular
engineering and
computer science,
Johns Hopkins University

Author information

Joshua Fern¹ and Rebecca Schulman^{1,2,*}

¹Chemical and Biomolecular Engineering, Johns Hopkins University, Baltimore, MD 21218;

²Computer Science, Johns Hopkins University, Baltimore, MD 21218

*E-mail: rschulm3@jhu.edu

References:

1. Qian, L. & Winfree, E. Scaling up digital circuit computation with DNA strand displacement cascades. *Science* **332**, 1196–1201 (2011).
2. Yao, D. *et al.* Integrating DNA-Strand-Displacement Circuitry with Self-Assembly of Spherical Nucleic Acids. *J. Am. Chem. Soc.* **137**, 14107–14113 (2015).
3. Zhang, D. Y., Turberfield, A. J., Yurke, B. & Winfree, E. Engineering entropy-driven reactions and networks catalyzed by DNA. *Science* **318**, 1121–1125 (2007).
4. Zhang, D. Y. & Winfree, E. Control of DNA Strand Displacement Kinetics Using Toehold Exchange. *J. Am. Chem. Soc.* **131**, 17303–17314 (2009).
5. Srinivas, N. *et al.* On the biophysics and kinetics of toehold-mediated DNA strand displacement. *Nucleic Acids Res.* **41**, 10641–10658 (2013).
6. Alberti, K. G. M. M. & Zimmet, P. Z. Definition, diagnosis and classification of diabetes mellitus and its complications. Part 1: diagnosis and classification of diabetes mellitus. Provisional report of a WHO Consultation. *Diabet. Med.* **15**, 539–553 (1998).
7. Mitchell, P. S. *et al.* Circulating microRNAs as stable blood-based markers for cancer detection. *Proc. Natl. Acad. Sci.* **105**, 10513–10518 (2008).
8. Bennett, B. D. *et al.* Absolute metabolite concentrations and implied enzyme active site occupancy in *Escherichia coli*. *Nat. Chem. Biol.* **5**, 593–599 (2009).

9. Seminario-Vidal, L., Lazarowski, E. R. & Okada, S. F. Assessment of Extracellular ATP Concentrations. in *Bioluminescence* (eds. Rich, P. B. & Douillet, C.) **574**, 25–36 (Humana Press, 2009).
10. Qian, L., Winfree, E. & Bruck, J. Neural network computation with DNA strand displacement cascades. *Nature* **475**, 368–372 (2011).
11. Nutiu, R. & Li, Y. Structure-switching signaling aptamers: transducing molecular recognition into fluorescence signaling. *Chem. - Eur. J.* **10**, 1868–1876 (2004).
12. Yin, P., Choi, H. M. T., Calvert, C. R. & Pierce, N. A. Programming biomolecular self-assembly pathways. *Nature* **451**, 318–322 (2008).
13. Chen, X., Briggs, N., McLain, J. R. & Ellington, A. D. Stacking nonenzymatic circuits for high signal gain. *Proc. Natl. Acad. Sci.* **110**, 5386–5391 (2013).
14. Gines, G. *et al.* Microscopic agents programmed by DNA circuits. *Nat. Nanotechnol.* **12**, 351–359 (2017).
15. Kotani, S. & Hughes, W. L. Multi-Arm Junctions for Dynamic DNA Nanotechnology. *J. Am. Chem. Soc.* **139**, 6363–6368 (2017).
16. Weitz, M. *et al.* Diversity in the dynamical behaviour of a compartmentalized programmable biochemical oscillator. *Nat. Chem.* **6**, 295–302 (2014).
17. Rammensee, S., Kang, M. S., Georgiou, K., Kumar, S. & Schaffer, D. V. Dynamics of Mechanosensitive Neural Stem Cell Differentiation. *Stem Cells Dayt. Ohio* **35**, 497–506 (2017).
18. Jiang, F. X., Yurke, B., Schloss, R. S., Firestein, B. L. & Langrana, N. A. The relationship between fibroblast growth and the dynamic stiffnesses of a DNA crosslinked hydrogel. *Biomaterials* **31**, 1199–1212 (2010).

19. Jiang, F. X., Yurke, B., Firestein, B. L. & Langrana, N. A. Neurite Outgrowth on a DNA Crosslinked Hydrogel with Tunable Stiffnesses. *Ann. Biomed. Eng.* **36**, 1565–1579 (2008).
20. Fern, J. & Schulman, R. Design and Characterization of DNA Strand-Displacement Circuits in Serum-Supplemented Cell Medium. *ACS Synth. Biol.* **6**, 1774–1783 (2017).
21. Groves, B. *et al.* Computing in mammalian cells with nucleic acid strand exchange. *Nat. Nanotechnol.* **11**, 287–294 (2015).
22. Cangialosi, A. *et al.* DNA sequence–directed shape change of photopatterned hydrogels via high-degree swelling. *Science* **357**, 1126–1130 (2017).

Reviewers' Comments:

Reviewer #1:

Remarks to the Author:

The authors have responded thoroughly to the previous referee reports. I recommend that the paper be published as is.

Reviewer #2:

Remarks to the Author:

All of my concerns have been addressed well, I recommend its acceptance by Nature Communications.

Reviewer #3:

Remarks to the Author:

The authors have done a good job in addressing the suggestions from the first round of reviews, including the addition of extra supplementary data to show the lack of cross-reactivity multiple hairpin systems with distinct sequence makeups, as well as a solid explanation of the issues relating to the K_d of the ATP aptasensor. As such I recommend acceptance of the revised manuscript.